# Research on the impact of User-Generated Content (UGC) in shaping the purchase behavior of environmentally friendly products and the moderatings role of brand reputation

Tu Ngoc Tran*, Nguyen Phan Thu Hang

Faculty of Business Administration, Saigon University, Ho Chi Minh City, Vietnam

* tntu@sgu.edu.vn

## Abstract

This paper aims to explore the effect of User-Generated Content (UGC) on the purchasing behavior of environmentally friendly products at Hospitality and Food Service Industry in Vietnam, particularly in Ho Chi Minh City. A conceptual model has been developed based on literature reviews and empirical studies. Furthermore, the Partial Least Squares Structural Equation Modeling (PLS-SEM) method was employed to investigate the impact of UGC on the purchasing behavior of environmentally friendly products at Hospitality and Food Service Industry in Vietnam. The results confirm that environmental concerns, attitudes, and the intention to purchase green products are all positively and significantly influenced by UGC. Especially, the results validate that brand reputation plays a moderating role in the connections between UGC and environmental concern, UGC and environmental attitude, as well as the relationship between the intention to purchase environmentally friendly products and actual purchasing behavior.

## Introduction

Environmental concerns are increasingly becoming a global issue, prompting researchers to examine the role of individuals in promoting green consumption behavior and environmental protection [131]. User-Generated Content (UGC) plays a prominent role in spreading environmentally friendly consumption behaviors. UGC includes posts, product reviews, videos, and content shared on social media, created and disseminated by consumers, helping to raise awareness and promote green actions on a global scale [34,35]. [49] defined environmental concern as the degree of human understanding of environmental issues, support for efforts to address them, and the willingness to contribute to solutions. Thanks to the widespread dissemination of UGC, awareness and information on environmental issues are quickly shared, making it easier for consumers to access knowledge about green consumption and

**Data availability statement:** The dataset generated and analyzed during this study is available in the Dryad data repository at: https://doi.org/10.5061/dryad.h44j0zq0c.

**Funding:** This research is funded by Saigon University, Ho Chi Minh City, Vietnam, under the Scientific Research Contract No. 369/HĐ-ĐHSG, dated May 5, 2025. Project code: CS.2025.A2.003 Project title: "The Impact of User-Generated Content (UGC) on the Purchase Behavior of Environmentally Friendly Products and the Moderating Role of Brand Reputation – Evidence from the Hospitality Industry in Vietnam". The funders had no role in study design, data collection and analysis, decision to publish, or preparation of the manuscript.

**Competing interests:** The authors have declared that no competing interests exist.

eco-friendly products [68]. The increase in UGC related to green products and services has provided a strong motivation for many consumers to shift their consumption behavior toward sustainability [72]. For instance, widely shared posts and videos on reducing plastic use or choosing eco-friendly products on social media platforms have positively influenced consumer awareness and actions.

The previous studies contribute a more coherent understanding of green purchase intentions and behaviors, such as [1,21,52,54,119,126]. These studies developed a model in which green value enhances green trust and purchase intention, respectively, and likewise, green risk attenuates it while acting as a mediator variable under the exploration [21]. Concretely, [1,126] found that based on media perception, positive influence is merely an unconscious physiological intention to purchase green products). They found that environmental concern and attitude positively impacted the intent to engage in green action. Srisathan et al. [119] investigated online green purchases of Thais and attitudes toward income.

Furthermore, in the Vietnamese context, [52] examined the growth of green consumption in Ho Chi Minh City during the COVID-19 pandemic due to health, which consumers must protect their health safely. Similarly, [54] identified environmental concerns and attitudes as the main drivers of green consumer behavior in Hue. Lastly, Tan [121]' study in Vietnam investigated the effect of eWOM on Vietnamese tourists and indicated that it is an important determinant in selecting environmentally eco-friendly destinations. So, they have a meaningful lens through which to make their business environment green and, simultaneously, understand consumer perspectives on how they can apply to their brand/products (and provide consumers with knowledge about green marketing segmentation). Those authors have also indicated that UGC raised consumer awareness of green products and has a strong impact on their purchasing intentions. Consumers often make purchasing decisions based on reviews and real-life experiences from other consumers, especially when it comes to environmentally friendly products [34,89]. According to the research by [35], when consumers share their experiences with green products on social platforms, this helps reinforce social norms regarding sustainable consumption and promotes the spread of green consumption habits. In Vietnam, the trend of consuming environmentally friendly products is growing rapidly, thanks to the development of UGC. Consumers are not only increasing their awareness but also sharing green actions, such as reducing plastic usage, choosing eco-friendly products, and adopting energy-saving measures on social media. Spending on green products has also risen significantly, not only due to government incentive policies but also because of the strong influence of UGC, which has expanded the market for these products.

In the study conducted by [42], the promotion of pro-environmental consumption activity is, in general, more efficient when it is planned and implemented systematically and evaluated in a continuous manner [78,128]. To do so, it is a fundamental requisite to clearly understand what environmentally sustainable consumer behavior is, to know the theories that are effective for explicating the behavior, to recognize the factors triggering it, and to know contemporary tourism and hospitality studies dealing with the behavior, which were not wholly uncovered in the extant literature.

To deepen the understanding of UGC's influence on consumer behavior, this study draws from Expectancy-Value Theory (EVT), Brand-Trust Theory (BTT), and Signaling Theory. These theories provide a robust framework for analyzing the moderating role of brand reputation in shaping the impact of UGC on environmental concerns, environmental attitudes, and purchase behaviors. First, Expectancy-Value Theory [32] posits that individuals make decisions based on the expected outcomes and perceived value of an action. In the context of green consumption, consumers engage with UGC to assess the perceived benefits of environmentally friendly products. Positive UGC reinforces consumer expectations of higher product quality, sustainability, and social responsibility, thereby increasing purchase intention and behavior [58,98]. Second, Brand-Trust Theory [84] highlights the role of trust in consumer decision-making, particularly in reducing perceived risks. When a hospitality business has a strong brand reputation, consumers exhibit greater confidence in the brand's sustainability claims, thereby reducing their reliance on UGC. Conversely, brands with weaker reputations see higher UGC influence on environmental attitudes and purchase intentions, as consumers seek social validation from peer reviews and testimonials [37].

Third, Signaling Theory [118] suggested that brand reputation acts as a signal of product quality and reliability, particularly in markets where information asymmetry exists [12]. In the hospitality and food service industry, businesses with established green credentials benefit from consumer trust, reducing the need for UGC to influence purchasing decisions. However, for lesser-known or emerging brands, UGC becomes a crucial factor in shaping consumer perceptions of environmental commitment [75,94].

Finally, the Theory of Planned Behaviour (TPB) has shown the ability to predict consumers' intention to purchase green products [59] and is widely used in the study of green consumption behavior, including sorting waste response, recycling e-waste or using energy from renewable sources [53,54,96]. Based on TPB, understanding the influence of UGC in social networks relates to attitude, subjective norm, and perceived behavioral control, which in turn affect green consumption intention. UGC contains positive information about green products and consumer experiences, such as sustainable consumption trends or personal stories highlighting green habits, which strongly impacts individuals' perception of sustainable consumption and leads to actual purchase shares.

While previous research on green consumption in Vietnam has largely focused on high-end dining establishments, this study broadens the scope to include casual dining restaurants, cafes, and food service businesses. The expansion acknowledges the diverse nature of consumer engagement with environmentally friendly products across different service formats [11,79]. UGC has proven instrumental in influencing purchase behavior across these settings, as consumers increasingly rely on digital content to assess restaurant sustainability practices, eco-friendly packaging, and ethical sourcing [91].

This broader scope allows for a more comprehensive analysis of how UGC interacts with brand reputation to shape environmental attitudes and behaviors across different segments of the hospitality industry. By examining a diverse range of food service businesses, this study provides valuable insights into the dynamic relationship between consumer-generated content, environmental consciousness, and sustainable purchasing behaviors.

Therefore, this study aims to investigate the impact of user-generated content on the purchasing behavior of eco-friendly products in Vietnam's hospitality and food service industry. Accordingly, the central research question is: "To what extent does User-Generated Content influence the purchasing behavior of eco-friendly products, considering the moderating role of Brand Reputation in Vietnam's Hospitality and Food Service Industry?".

## Literature review and hyposthesis development

### Literature review

**User-generated content (UGC) and consumer behavior.** User-generated content (UGC), also known as electronic word of mouth (eWOM), is a form of traditional word of mouth setting in the online context but differs from other forms in

that it is unsolicited and voluntary cooperation from volunteers sharing information which can provide other orientating options [77]. It is made up of thoughts and emotions of online users about products or services based on their experience. According to [6], users shared that information via social media, assuming it may guide others as a reference to perform better purchase intentions. Moreover, Sethna et al. [111] stated that UGC is those reviews where another user shares a say on a product, like any comments or criticisms based on product usage, packaging or design, usability, and delivery.

Particularly, UGC plays a significant role in shaping how potential customers search for restaurants, especially by focusing on restaurant dishes [91]. It also influences their destination choices and purchasing decisions [11,79,124]. Moreover, UGC helps differentiate restaurants in competitive markets through eWOM [106], where positive online reviews enhance brand reputation by highlighting service and food quality [9,36,91]. As [91] note, monitoring online feedback is essential for restaurants [93] because social networks allow consumers to share experiences, often driven by altruism, which strengthens the impact of UGC through psychological and social connections [61,138].

This consumer engagement with UGC aligns with established consumer behavior theories, which emphasize the decision-making process and the situational factors influencing it [33]. According to [65], the consumer behavior model identifies key attributes that shape purchasing decisions, including first-generation stimuli such as marketing efforts, situational factors, and buyer characteristics. These elements collectively mold the decision-making process and influence purchase choices. In the context of UGC, the type of content, the digital environment, and consumer attributes interact to determine how online reviews, recommendations, and discussions drive behavior when individuals evaluate and choose products or services.

As defined by [57], consumer behavior involves the acquisition, consumption, and disposal of goods, services, and ideas by individuals or groups. This process is universal and pervasive, influencing decisions in daily life across all societies and cultures. Recent research has shifted to a decision-making perspective, particularly in sustainable consumer behavior, where motivations and psychological factors drive environmentally impactful behaviors are examined [127]. Environmentally sustainable consumer behavior is critical for environmental protection, benefiting society [41,42]. Green products, their pricing [82,112], and effective communication of consumer beliefs regarding these items are increasingly important [92]. As a result, ensuring that information about green products is accessible and understandable is essential [80]. Moreover, technological adaptation in trading systems has facilitated online purchasing, expanded consumer reach, and influenced behavior [100].

Sustainability messaging was proved as a drivers of customers' eco-friendly behavior in recent research such as such as Avrami et al. [4]; White et al. [133]. Additionally, there is a positive relationship between pro-environmental attitude and environmentally sustainable product or service behavior that's mentioned in the study of De Canio et al. [26]. In the context of the hotel and food service industry, especially in emerging economies, UGC plays an important role in shaping customer perceptions and decisions. Many studies have demonstrated that online reviews, images posted and shared by customers on social media have a significant impact on the intention to choose a restaurant or hotel [73,95,105]. In developing countries like Vietnam, according to [66,109], UGC becomes a reliable source of information, helping users minimize risks in the purchase decision process because consumers do not fully trust information created by traditional media channels. Particularly, in the food/food industry, customer-posted reviews, experiences, and photos are a trusted source of information from the community, helping consumers evaluate the value of services and verify businesses' "green" commitments by increasing transparency, authenticity, and trust [10]. Therefore, to clarify the role of cultural factors, brand trust, and sustainable consumption motivation in consumer decision-making, it is necessary to study the impact of UGC on green products and services purchase behavior in emerging markets such as Vietnam.

**Theory of Planned Behavior (TBP).** The Theory of Planned Behavior (TPB) posits that human behavior is predicted by behavioral intention, which in turn is determined by three constructs: attitude towards the behavior, subjective norms and perceived behavioral control (PBC) [3]. In other words, what makes TPB different than the TRA is PBC – an individual's belief in their capacity to perform a behavior given available resources and opportunities.

According to TPB, behavioral intention is the main predictor of an individual's actual behavior and perception of control. An individual's actual behavior and perception of control strongly influence the consumer's future actions. TPB provides a valuable framework for understanding consumer behavior considered in the restaurant context, especially when considering factors that influence consumer decisions, as shown by [20,74,114]. Specifically, according to Lombardi et al. [74], consumer behavior theory is essential to identifying factors that enhance or limit consumers' interest in local or sustainable food options. The TPB has been widely applied in research on consumer intentions, such as visiting green hotels [20] and choosing restaurants with organic menus [114]. It also helps explain the psychosocial and demographic factors that shape the intention to purchase sustainably produced food [107], making it highly relevant to understanding consumer behavior in the restaurant context [14].

In this study, the author does not apply the entire TPB framework but only inherits a part, specifically the component "attitudes towards behavior", which is the central element in the TPB model. This component explains how UGC influences consumers' attitudes toward green products, thereby shaping their intention and behaviour to purchase environmentally friendly products.

Emphasizing the "attitude" factor has a scientific basis, because studies on green consumer behavior show that this is an important premise in forming behavioral intentions [43,114]. It is believed that positive attitudes towards the environment are expected to promote the perception that choosing green products is the right action, stemming from moral values and social benefits [20,97]. This is consistent with TPB theory, which suggests that positive perceptions are closely related to behavioral intentions.

In addition to TPB, this study also incorporates Signaling Theory [118] and Brand-Trust Theory [84] to create a more reliable theoretical foundation. While TPB explains the mechanism of behavioral intention formation through attitude, Signaling Theory explains how online signals such as UGC help consumers evaluate the authenticity of information about green products [28,134]. At the same time, Brand-Trust Theory introduces the concept of brand trust, demonstrating that trust plays a moderating role in the relationship between attitude and green behavioral intention [19,27]. Based on this integration, the research model achieves higher generalizability and overcomes the limitation of only relying on a part of TPB.

**Signaling theory.** Signaling theory [118] posits that in markets characterized by information asymmetry, consumers rely on observable signals such as brand reputation to assess product credibility and reduce uncertainty. This is particularly relevant in the evaluation of sustainability claims, where consumers may struggle to verify the authenticity of a brand's environmental commitments. In the context of UGC and environmental concerns, brand reputation functions as a moderating factor that influences how consumers process information. A well-established brand with a strong reputation serves as a credible signal of environmental responsibility, diminishing consumers' reliance on UGC when forming perceptions about a brand's sustainability efforts [28]. Consequently, for reputable brands, UGC may have a weaker effect on shaping environmental concerns, as consumers already trust the brand's commitments. However, for lesser-known brands or those with weaker reputations, UGC becomes a crucial determinant in shaping consumers' environmental concerns, as it provides additional credibility and social proof [134]. This dynamic highlights how signaling theory explains variations in UGC's impact, emphasizing the role of brand reputation in moderating consumer responses to sustainability-related messages.

**Expectancy-Value Theory (EVT).** Expectancy-Value Theory (EVT) [32] posits that individuals' purchase behaviors are guided by their expectations of a product's benefits and the value they assign to those benefits. In the context of purchasing environmentally friendly products, consumers develop purchase intentions based on their perceptions of environmental benefits and their personal commitment to sustainability [50]. However, brand reputation moderates the transition from intention to actual behavior by shaping consumers' trust in a brand's environmental claims. A strong brand reputation serves as a credibility signal, reinforcing consumer confidence and reducing uncertainty, thereby strengthening the intention-behavior relationship [28]. When consumers perceive a brand as trustworthy in its sustainability efforts,

they are more likely to follow through with their purchase intentions [19]. Conversely, for brands with weaker reputations, skepticism may arise, weakening the link between intention and behavior as consumers seek additional validation before making a purchase decision [102]. Thus, EVT provides a robust framework for understanding how brand reputation influences consumers' decision-making process in purchasing environmentally friendly products.

Conversely, for brands with weaker reputations, the gap between purchase intention and actual behavior may widen due to skepticism regarding the authenticity of their environmental claims [102]. Consumers may intend to buy environmentally friendly products but hesitate to act due to concerns about greenwashing or misleading sustainability claims [69]. In such cases, brand reputation plays a crucial role in either reinforcing or undermining consumers' motivation to follow through with their intentions. The theory thus provides a valuable framework for understanding how brand reputation moderates the intention-behavior relationship in the context of environmentally friendly purchases. A well-established reputation strengthens this link by reducing uncertainty and enhancing trust, whereas a weak reputation increases reliance on external validation, such as third-party certifications or peer recommendations, before making a purchase decision.

**Brand-trust theory.** Brand-Trust Theory [84] posits that trust is a fundamental factor in fostering strong relationships between consumers and brands, influencing consumer attitudes and behaviors. Trust is built over time through consistent brand performance, credibility, and perceived reliability. In the context of UGC and environmental attitudes, brand reputation – rooted in trust – plays a crucial moderating role in shaping how consumers interpret and respond to UGC. A brand with a strong, trusted reputation reduces uncertainty and reinforces positive consumer perceptions about its environmental commitments [19]. As a result, consumers who already trust a brand may rely less on UGC to form their environmental attitudes, as they consider the brand's sustainability efforts to be credible and well-established.

Conversely, for brands with lower trust or weaker reputations, UGC serves as a more influential source of information in shaping consumers' environmental attitudes. When a brand lacks a strong trust foundation, consumers may depend more on peer-generated content, such as reviews and social media discussions, to assess its environmental responsibility [115]. This dynamic aligns with Brand-Trust Theory, which highlights that trust mitigates uncertainty and enhances consumer confidence in a brand's claims [27]. Thus, when brand reputation is strong, the moderating role of UGC in shaping environmental attitudes weakens, whereas in cases where brand trust is lacking, UGC becomes a critical determinant of consumer perception. This perspective underscores the interplay between brand trust, reputation, and UGC in influencing consumer attitudes toward sustainability.

## Empirical studies

Y.-S. Chen & Chang [21] have performed a theoretical approach to observe green value and green risk effects on green purchase intention with the mediator of green trust. The results show that green value has a significant positive effect on both green trust and purchase intention, but green risk negatively affects these constructs. The above results suggested that green trust was an intermediary in the relationship between green purchase intention and its antecedents. Theoretically, increasing green value and reducing green risk leads to higher levels of trust and the betterment of the environmental goods in which such trust is a necessary antecedent for green purchase intentions.

Trivedi et al. [126] analyzed the impact of media on environmental attitudes, green purchase intention, and behavior. The study provided a streamlined theoretical framework that included the role of media, attitudes toward eco-friendly packaging, ecological concern, and consumption efficacy. Their results indicated that intrinsic environmental attitudes and attitudes toward green packaging significantly shaped green purchase intentions, while extrinsic environmental attitudes did not have a significant effect.

Afridi et al. [1] investigated the role of generativity in green purchasing behavior and considered the mediating role of environmental concern and pro-social attitudes. The results showed that environmental concern positively and significantly affected green purchasing behavior when consumers are concerned about current environmental issues. Pro-social attitudes also had a crucial impact, representing how individuals treat society and others.

Srisathan et al. [119] studied the green awakening attitudes of customers toward online platforms in emerging economies, focusing on age, gender, and income differences in Thailand. Results indicated that customers' attitudes toward purchasing green products online are explained by relative advantage, online social norms, and perceived risk. In contrast, online compatibility did not have a significant effect.

In Vietnam, Ho et al. [52] examined the change in consumer perceptions of health and risk due to the COVID-19 pandemic, which increased green product consumption. Their results showed that green product knowledge, orientation, and social influence positively affected attitudes toward green product purchasing, while fear of COVID-19 and green shopping attitudes also influenced purchase intentions. Knowledge and social influence had indirect effects on green shopping attitudes. The research provided valuable insights for green businesses in promoting consumer attitudes and purchase intentions post-pandemic. Whereas Hoang et al. [54] studied the determinants of green consumer behavior in another city in Vietnam, Hue. This study employed the SEM model, showing that attitudes to greenwashing and environmental concerns impact consumer decision-making processes, purchase intention, and behavior.

Tan [121] assessed the impact of eWOM on Vietnamese tourists' tendency to choose environmentally friendly destinations. Based on the previously presented eWOM theories, this study attempts to establish and show that attitude, subjective norm, and environmental concern travel motivation together explain the intention to select green destinations. The study offered unique insights for Vietnamese tourism organizations and marketers consisting of components used in green tourism promotion.

*In conclusion, while existing research explores factors like trust, environmental concern, media, and attitudes influencing green purchasing, this study uniquely focuses on how online consumer content shapes purchase behavior at Hospitality and Food Service Industry in Vietnam. It offers valuable insights into the moderator roles of brannd reputation in the relationship between UGC and environmental concern; UGC and environmental attitude; Intention to purchase environmentally friendly products and Purchasing Behavior.*

## Hypothesis development

### For the direct effects.

**The relationship between UGC and environmental concerns and attitudes:** According to Hadamitzki [39], online trends related to UGC have been studied in various contexts, especially in the restaurant industry, where businesses integrate these trends into their operations. UGC includes online content such as customer testimonials [8], which contribute to the impact of eWOM, enhance consumer engagement [24], and improve their attitudes toward the brand [110]. UGC has increasingly played a crucial role [5] and is highly valued by online consumers when seeking information about products and services [103].

In another descriptive study, [110] research results add online marketing and social media impact, which affects consumers' brand decisions and perceptions, have also been recognized by Oliveira & Casais [91]. These actions have led to the rise of eWOM [73], influencing the purchasing decisions of potential consumers [113] while also turning them into content producers (UGC) as they share experiences and provide recommendations on products and services such as hotels and restaurants [16,38]. [76] conducted a study in the restaurant industry arena, which suggests that restaurant chains are typically the least sustainable when it comes to environmental policy because their broad implementation policies have historically been more complex and less willing to change at each franchisee [108]. The restaurant industry is one of the largest and most profitable. However, at the same time, it has a considerable pollution load: such significance lies in the emissions and waste generation [62], as well as a significant water consumption effect and energy consumption in general. This shows the sensitivity and dedication of people or groups to the environment. Sustainability and environmental awareness are increasingly unavoidable global issues connected to responsible environmental practices.

UGC strongly influences consumers' attitudes about restaurants, mainly through social media and online reviews [62]. When judging an aspect of a restaurant from the perspective of consumers, it is easier to be swayed by eWOM and other

types of electronic sources [36,103] due to its rank in review sites or positive comments. In recent years, social media has been used to disclose experiences related to eating and drinking; where some consumers visit restaurants inspired by the opinion of others, and many others only realize the importance of doing so after their meals have taken place [91,138]. In summary, according to Sultan et al. [120], UGC in social media can amplify individual perceptions, enhancing their sensitivity to environmental issues.

While, customer attitudes, defined as overall evaluations, are influenced by motivations, values, and beliefs [90,116]. Based on the study conducted by Sultan et al. [120], UGC influences the formation of positive environmental attitudes by fostering emotional engagement with environmental issues. The more individuals are exposed to environmentally oriented UGC, the more their attitudes align with responsible environmental behavior. In the restaurant sector, attitudes are shaped by factors such as food quality, service, ambiance, and menu, which impact loyalty and the likelihood of recommendations [60,93]. Trust in online reviews and restaurant rankings also has a significant impact on customer attitudes [103]. There is a close connection between UGC and customer attitudes. Consumers use UGC to evaluate restaurants and contribute their feedback, creating a continuous cycle that reinforces the restaurant's reputation and influences consumer behavior [61,120,138].

Based on the above agruments, the authors proposed the hypotheses are:

*Hypothesis 1: UGC positively impacts environmental concerns at Hospitality and Food Service Industry in Vietnam.*

*Hypothesis 2: UGC positively influences environmental attitude at Hospitality and Food Service Industry in Vietnam.*

**The relationship between UGC and purchasing behavior for environmentally friendly products:** UGC plays a vital role in the era of digitalization, considerably affecting consumer behavior by representing beliefs, attitudes, and behavior [44,117]. In modern times, consumers depend on UGC through media like Facebook, Instagram, and Zalo… to have more information about the product and feedback from other customers [6,13]. While UGC, often in the form of reviews, images, and videos, aids consumers in making informed decisions, the overwhelming content can complicate the decision-making process [55].

Pro-environmental behavior encompasses actions by individuals or groups aimed at promoting the sustainable use of natural resources [44,104]. In the restaurant industry, it is still "far from sustainable" due to food waste, plastic waste, and emissions, along with high consumption of water and energy [62]. Environmental issues must be considered because of their impact on the environment, as mentioned above. Therefore, efforts to understand and predict consumers' environmental behavior have generated a large body of literature, much of which investigates these issues based on the assumption that behavior results from a linear and rational decision-making process [46].

Bahtar & Muda [6]; Horst et al. [55]; Racherla & Friske [103] have demonstrated that UGC exerts a positive influence on purchase decisions. Notably, the research conducted by [138] highlighted this relationship specifically within the context of the restaurant sector. As a result, we propose the hypothesis that *UGC positively influences purchasing behavior for environmentally friendly products at Hospitality and Food Service Industry in Vietnam (Hypothesis 3).*

**The relationship between environmental concerns, purchase intentions, and purchasing behavior for environmentally friendly products:** According to Balaskas et al. [7], environmental concern explains green consumer behavior; it pertains to the psychological and behavioral processes through which human beings become proponents of eco-friendly purchasing decisions or actively advocate for the environment. This concern is also closely linked to consumers' tendency to join environmental support groups. It measures, theoretically rather crudely, the level of environmental concern and awareness as well as a commitment to reducing environmental problems and support for conservation [85]. People act environmentally protective because they use pro-environmental behavior as a proxy to satisfy those desires and intentions of protecting nature or the environment [81,101].

Environmental concern reflects consumers' general attitude toward environmental protection and plays a significant role in sustainable consumption [26,132]. It is a factor influencing consumers' motivation to adopt a sustainable lifestyle

[87,132]. Hartmann & Apaolaza-Ibáñez [47] demonstrated that environmental concern directly affects the intention to purchase green brands/products. This concern is also a key driver of sustainable food purchase intentions, as environmentally conscious consumers choose products with lower environmental impacts [26,45,123]. Recent studies further indicate that higher sensitivity to environmental issues can increase the intention to buy organic food [15,122].

Environmental concern reflects consumers' general attitude toward environmental protection [26,132] and plays a significant role in sustainable consumption behavior. It is a determining factor influencing consumers' motivation to adopt a sustainable lifestyle [87,132]. Hartmann & Apaolaza-Ibáñez [47] demonstrated that environmental concern directly affects the intention to purchase green brands/products. This concern is also a key driver of sustainable food purchase intentions, as environmentally conscious consumers tend to choose products with lower environmental impacts [26,45,123]. Recent studies further indicate that higher sensitivity to environmental issues can increase the intention to buy organic food [15,122].

In the work of Kim & Choi [64], they identified a significant positive relationship between green purchasing behavior and concern for the environment. Another important issue in India is the new generation of young consumers who, being well-aware and concerned for the environment [137], also tend to make purchases accordingly. Furthermore, purchasing decisions are explained by environmental concerns, analyzed in the studies by Balaskas et al. [7]; Mostafa [86].

*Hypothesis H4: Environmental concerns positively influence the intention to purchase environmentally friendly products at Hospitality and Food Service Industry in Vietnam.*

*Hypothesis H5: Environmental concerns positively influence the behavior of purchasing environmentally friendly products at Hospitality and Food Service Industry in Vietnam.*

**Relationship between purchase intentions and purchasing behavior for environmentally friendly products:** According to Chan [17], a consumer's intention to purchase green products can be assessed by evaluating factors such as buying eco-friendly products, switching to environmentally friendly brands, and choosing green versions of products [17]. Current and future consumer purchasing decisions for environmentally friendly products are often evaluated through the intention to buy green products. Both external and internal factors interact, creating a certain gap between intention and the actual behavior of purchasing eco-friendly products [48,71].

Studies on green purchasing behavior have shown a strong positive relationship between purchase intention and purchasing behavior [54]. Furthermore, research results also indicate that purchase intention has the strongest and most direct influence on customers' purchasing behavior [124].

In the case of the Hospitality and Food Service Industry in Vietnam, purchase intention for environmentally friendly products will directly impact purchase behavior. Similarly, consumers with high levels of CSR consciousness are more likely to adopt sustainable buying decisions, choosing to favor services and products that demonstrate a staunch commitment to environmental conservation [21]. Given the arguments above, the following hypothesis is proposed.:

*Hypothesis 6 (H6): Purchase intention for environmentally friendly products positively influences purchasing behavior at Hospitality and Food Service Industry in Vietnam.*

**The moderating role of brand reputation.**

**In the relationship between UGC and environmental concerns:** Signaling theory [118] explains how companies use trust signals, such as brand equity and CSR initiatives, to establish credibility. Well-known brands accumulate these signals over time, reducing consumer reliance on UGC to assess their environmental commitment. In contrast, lesser-known brands lack strong credibility signals, making consumers more dependent on UGC for evaluating sustainability claims. Therefore, brand reputation is expected to moderate the relationship between UGC and consumer environmental concern.

Borah & Tellis [12] found that UGC has a weaker impact on high-reputation brands, as their established credibility reduces the need for external validation. In contrast, [75] demonstrated that for low-reputation brands, UGC becomes a

crucial factor in shaping consumer perceptions, effectively substituting for brand credibility. As a result, consumers rely heavily on positive UGC to assess the sincerity of these brands' environmental efforts. This is especially proper in the hospitality industry, where it is challenging for customers to directly assess "green" factors, such as the sources of raw materials or operating procedures. For highly reputable brands, the brand name and image reduce the need for verification through UGC. Conversely, for less well-known brands, UGC becomes an objective confirmation channel for consumers to evaluate environmental commitment and service quality.

The effect of negative UGC further highlights this dynamic demonstrated by [136]. The authors found that strong brands are more resilient to negative UGC, while weaker brands are highly vulnerable. Without an established reputation, negative UGC can increase skepticism about their environmental claims and raise concerns about greenwashing.

Building on these arguments, this study proposes the following hypothesis:

*H7: Brand reputation moderates the relationship between UGC and consumer environmental concerns at hospitality and food service industry in Vietnam.*

**In the relationship between UGC and environmental attitude:** As mentioned above, Brand-Trust Theory [84] explains how brand trust shapes consumer responses to external information. When a brand has a strong reputation, consumers are more likely to trust its environmental commitment, making UGC less influential in shaping environmental attitudes. Conversely, when a brand has a weak reputation, consumers are uncertain about its sustainability efforts, leading them to rely more on UGC to assess its environmental credibility. This suggests that brand reputation moderates the effect of UGC on environmental attitudes.

Traore [125] found that consumers depend less on UGC when engaging with well-known sustainable brands, as their credibility is already established. However, for lesser-known brands, UGC plays a critical role in shaping perceptions of sustainability. Similarly, [37] showed that low-reputation brands benefit more from UGC engagement than high-reputation brands in the sustainability domain, as consumers use UGC to validate their environmental claims.

In the hospitality and food industry, brand reputation is not only the public image of the business but also the commitment to service quality, food safety and sustainable development orientation [83]. A strong brand helps customers feel more secure and confident, reducing concerns about risks when choosing a service. [29] argue that when consumers are familiar with a brand, they see it as a guarantee of quality, which reduces the need to search for further information. Therefore, in the case of strong brands, UGC often only plays a role in reinforcing trust rather than being a determinant of choice behavior, because brand reputation is enough to convince customers [70,130]. In contrast, for low-reputation or new brands, consumers lack the basis to evaluate the reliability of brand information, so they tend to rely more on other people's reviews and real-life experiences on social networks (online reviews, ratings, feedback). The results are supported by Signaling Theory and Brand Trust Theory, as well as empirical studies by [31,136]. Accordingly, UGC has a stronger impact on new or less well-known brands, because consumers rely on others' reviews to assess trustworthiness and reduce risk. In contrast, strong brands build trust through accumulated reputation, while weak brands rely on UGC to reinforce trust [22,63].

*H8: Brand reputation moderates the relationship between UGC and consumer environmental attitude at hospitality and food service industry in Vietnam.*

**In the relationship between Intention to purchase environmentally friendly products and purchasing behavior:** As mentioned above, EVT explains that individuals make decisions based on expected benefits and the subjective value they assign to them. In consumer behavior, EVT suggests that purchase intentions depend on expectancy – beliefs about whether a product will deliver the expected benefits, such as environmental sustainability – and value, or the perceived importance of those benefits, including quality, brand reputation, and authenticity. When purchasing environmentally friendly products, consumers weigh both environmental impact and brand credibility. A strong brand reputation

enhances trust and perceived product effectiveness, increasing the likelihood of purchase, while a weak reputation fosters skepticism, potentially hindering this process.

Empirical studies have consistently demonstrated that brand reputation significantly influences the relationship between consumers' purchase intentions and their actual purchasing behaviors. A strong brand reputation enhances trust and reduces perceived risk, thereby strengthening the intention-behavior link. For instance, [88] found that consumers were more likely to act on their eco-friendly purchase intentions when dealing with reputable brands.

Conversely, a weak brand reputation can lead to purchase hesitation. Vermeir & Verbeke [129] observed that even when consumers had strong purchase intentions, they often refrained from purchasing if the brand's credibility was low. Trust deficits and perceived risks acted as barriers, preventing intentions from converting into actions.

Moreover, brand credibility plays a crucial role in mitigating the negative effects of greenwashing mentioned by [99]. Their findings found that consumers were more likely to delay or cancel purchases when a brand had a history of misleading environmental claims, further weakening the intention-behavior relationship. Consumers rely on brand reputation to assess credibility, and when trust is compromised, skepticism reduces their willingness to act on their initial purchase intentions.

Additional research supports these findings. A study on the impact of corporate reputation on brand attitude and purchase intention revealed that brand awareness and perceived quality significantly affect purchase decisions. Similarly, research on the mechanism of brand reputation on consumer purchase intention highlighted that a strong brand reputation positively influences consumer behavior by enhancing trust and perceived value [135].

Put in another way, [67] identified that brand reputation plays a crucial role in assisting customers with their purchasing decisions and strengthening their trust. [2] emphasized that when a company has a strong brand reputation, customers are more likely to trust its products and feel a sense of satisfaction and pride in their purchase decisions. For businesses, a positive brand reputation contributes favorably to financial performance [51], equity [30], and profits [18]. [23] suggested that to establish a strong brand reputation, companies must demonstrate social responsibility.

*H8: Brand reputation moderates the relationship between Intention to purchase environmentally friendly products and Purchasing Behavior at Hospitality and Food Service Industry in Vietnam.*

## Research model and methodology

### Research model

Fig 1 presents the proposed research model. The model illustrates how User-Generated Content (UGC) influences consumers' environmental concerns and environmental attitudes, which in turn shape their intention to purchase environmentally friendly products and their actual purchasing behavior. Additionally, brand reputation is introduced as a moderating variable, reinforcing or weakening the relationships between environmental concerns, purchase intention, and purchasing behavior. This framework integrates the cognitive and affective mechanisms through which UGC contributes to sustainable consumer behavior in the hospitality industry.

### Research methods

This study employs a quantitative approach using Partial Least Squares Structural Equation Modeling (PLS-SEM) to address the research question. Data collected is processed through Smart PLS software, following a structured analytical process: descriptive analysis, factor loading and convergent validity assessment, reliability testing of scales, discriminant validity evaluation, and multicollinearity assessment. Additionally, model fit is tested using $R^2$ or adjusted $R^2$ values.

The PLS-SEM method was chosen because the research model is exploratory, includes both mediating and moderating variables, and focuses on predicting the relationship between latent variables. This method does not require data to follow a normal distribution, is suitable for small to medium sample sizes, and allows for the simultaneous assessment of

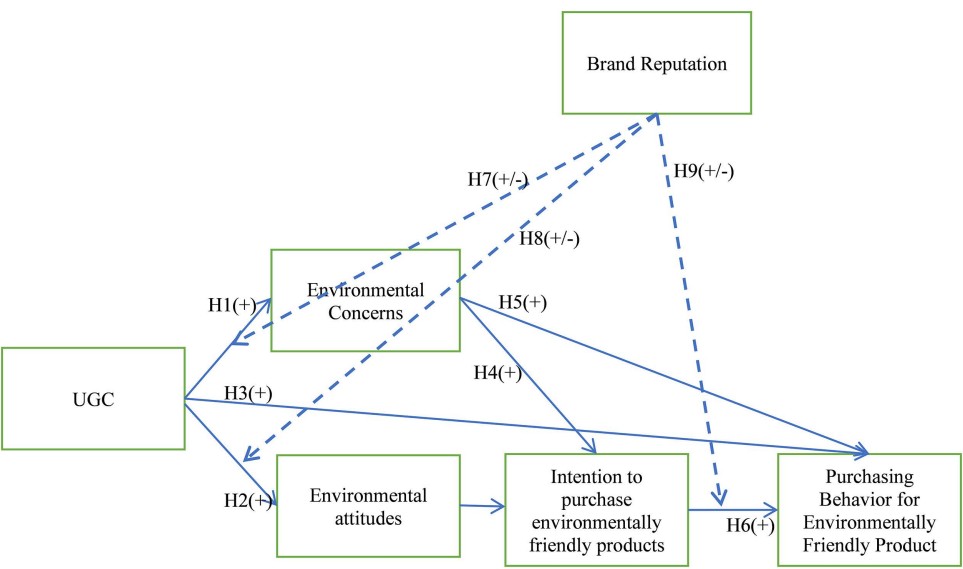

**Fig 1. Research model.**

direct, indirect, and moderating effects. The research model includes four main groups of factors: (1) UGC as the independent factor, (2) Environmental Attitude, concerns as the mediating factors, (3) Brand Reputation acts as a moderator, and (4) Purchasing Behavior of Environmentally Friendly Products as the dependent factor. The use of PLS-SEM enables practical testing of the influence mechanism of UGC on the purchasing behavior of environmentally friendly products in the hotel and catering industry.

Besides, the f-square ($f^2$) statistic in SmartPLS is a key effect size measure that assesses the impact of an independent variable on a dependent variable in a PLS-SEM framework. It helps determine whether an exogenous construct has a substantive impact on the endogenous construct. According to [25], the effect size values are interpreted as follows: Small effect: if $f^2 \geq 0.02$; Medium effect: if $f^2 \geq 0.15$; Large effect: if $f^2 \geq 0.35$.

## Data collection and sample

The study employed a random sampling questionnaire method to collect data, measured at an ordinal level using a Likert scale. This approach allowed respondents to indicate their level of agreement or disagreement with statements related to the research topic. Data collection took place over a four-month period, from January 5, 2025, to March 05, 2025. Participants were recruited from the Hospitality and Food Service Industry in Vietnam, specifically from five-star restaurants in Ho Chi Minh City, including Elson Meat & Wine, Sorae Restaurant – Lounge, Moo Beef Steak Prime, The Log – Dine & Wine, and Ussina Hotel. Conducting surveys with customers at high-end restaurants helps ensure uniformity in service quality, experience, and awareness levels related to sustainable consumption. However, the authors found that the survey scope at 5-star restaurants in Ho Chi Minh City would result in a research sample that does not fully represent the characteristics of all Vietnamese consumers, thereby limiting the generalizability of the research results. Therefore, in subsequent studies, it is necessary to expand the scope of data collection to include different types of dining establishments (such as 3–4-star restaurants, cafes, or green dining chains) and more diverse geographical areas across the country to improve the generalizability and applicability of the results.

Participation in the study was entirely voluntary, and all participants provided verbally informed consent before taking part. Prior to responding, participants were fully informed about the study's objectives, the scope of their participation, and

their right to withdraw at any time without consequences. Verbal consent was documented by the survey administrators before data collection commenced. Among the participants, two respondents were under the age of 18. For these cases, parental or guardian consent was obtained prior to data collection. The study was approved by the Ethics Committee of Saigon University, which confirmed that the research procedures complied with institutional ethical standards and national regulations for human participant research.

A total of 350 responses were initially collected, but after removing eight incomplete responses, the final dataset consisted of 336 valid responses, resulting in a final response rate of 96%. This sample size exceeds the five-times-the-number-of-items criterion established by Bollen (1989), ensuring its adequacy for analysis.

## Research results and discussion

### Descriptive statistics

The demographic characteristics of participants are provided in Table 1 (with a sample size of 336). The analysis reveals 336 respondents in the sample; 48.8% are male (164 individuals) and 51.2% female (172 individuals). By age distribution, the biggest group, representing 37.5% of samples (126 respondents), were people aged 31–45, followed by those aged 46–55, representing a quarter (26.5%; 89 respondents). Those 55 and older comprised 22% of respondents (74 people), and those aged 19–30, 13.4% (45 respondents). Lastly, 0.6% of the population studied comprises only two individuals below 18. It shows a predominantly middle-aged survey profile, with only a marginal female lead.

### Research results

Table 2 presents Cronbach's alpha, composite reliability, and average variance extracted (AVE) to assess the reliability and validity of the latent constructs in the model. Cronbach's alpha values for all constructs exceed 0.9, indicating excellent internal consistency. Similarly, the composite reliability values for all constructs are above 0.9, reaffirming the strong internal consistency: HVM (0.989), LN (0.989), TD (0.966), UGC (0.993), YD (0.991), and BR (0.992). The AVE values, exceeding 0.8, confirm convergent validity, as they surpass the acceptable threshold of 0.5. Specifically, HVM (0.956), LN (0.956), TD (0.876), UGC (0.974), YD (0.966), and BR (0.976) reflect that these constructs capture a significant portion of the variance, further validating the robustness of the measurement model.

Table 3 concerns discriminant validity, where the Heterotrait-Monotrait ratio of correlations (HTMT) between all constructs is measured to assess discriminant validity. A value of less than 0.85 is usually considered adequate, demonstrating perfect divergent validity. The construct relationships are shown in Table 3 satified the rule. It means that the

**Table 1. Demographic analysis.**

| Index | Frequency | Percentage (%) |
|---|---|---|
| 1. Gender | | |
| Male | 164 | 48.8% |
| Female | 172 | 51.2% |
| **Total** | **336** | **100%** |
| 2. Age | | |
| Under 18 | 2 | 0.6% |
| 19-30 | 45 | 13.4% |
| 31-45 | 126 | 37.5% |
| 46-55 | 89 | 26.5% |
| Over 55 | 74 | 22.0% |
| **Total** | **336** | **100%** |

**Table 2. Composite reliability.**

|  | Cronbach's alpha | Composite reliability (rho_c) | Average variance extracted (AVE) |
|---|---|---|---|
| BR | 0.988 | 0.992 | 0.976 |
| HVM | 0.985 | 0.989 | 0.956 |
| LN | 0.985 | 0.989 | 0.956 |
| TD | 0.953 | 0.966 | 0.876 |
| UGC | 0.991 | 0.993 | 0.974 |
| YD | 0.988 | 0.991 | 0.966 |

*Notes*: BR is brand reputation; HVM is purchasing behavior for environmentally friendly products; LN is Environmental concerns; TD is Environmental Attitude; UGC is User-generated content; YD is Intention to purchase environmentally friendly products.

**Table 3. Discriminant validity assessment using Heterotrait-monotrait ratio (HTMT).**

|  | BR | HVM | LN | TD | UGC | YD | BR x UGC | BR x YD |
|---|---|---|---|---|---|---|---|---|
| BR |  |  |  |  |  |  |  |  |
| HVM | 0.436 |  |  |  |  |  |  |  |
| LN | 0.259 | 0.481 |  |  |  |  |  |  |
| TD | 0.258 | 0.348 | 0.488 |  |  |  |  |  |
| UGC | 0.191 | 0.494 | 0.688 | 0.475 |  |  |  |  |
| YD | 0.374 | 0.502 | 0.567 | 0.531 | 0.624 |  |  |  |
| BR x UGC | 0.047 | 0.036 | 0.344 | 0.308 | 0.276 | 0.232 |  |  |
| BR x YD | 0.131 | 0.029 | 0.205 | 0.243 | 0.187 | 0.475 | 0.648 |  |

*Notes*: BR is brand reputation; HVM is purchasing behavior for environmentally friendly products; LN is Environmental concerns; TD is Environmental Attitude; UGC is User-generated content; YD is Intention to purchase environmentally friendly products.

constructs are diversified enough to be distinguishable from one another. Consequently, based on the research from Table 3, it may be inferred that the constructs included in the study are distinct and independent, providing further reliability to the measure.

Fig 2 illustrates Structural equation modeling (PLS- SEM) by using a 5,000 resample bootstrapping procedure conducted through SmartPLS. The bootstrapping aims to assess the reliability and stability of the estimated path coefficients. It allows the calculation of standard errors and t-values, which are crucial for hypothesis testing as proposed in section of Hypothesis development.

Table 4 presents the VIF values used to evaluate multicollinearity. Hair et al. [40] state that acceptable VIF values are generally below 3, with values closer to 1 indicating no multicollinearity. In particular, the analysis is as follows. The obtained results indicate all VIF values satified the rule of thumbs set by Hair et al. [40], thus, there is no multicollinearity in the model.

R-squared from Table 5 measures the proportion of variance in the dependent variable that is explained by the independent variables in a model. It indicates how well the data fits the model.

First, for HVM: R-squared is 0.452, meaning that 45.2% of the variance in HVM is explained by the independent variables. This suggests a relatively good fit for the model in explaining the variability of HVM. While the adjusted R-squared is 0.443, very close to the R-squared value, which indicates that the model is well-specified without significant overfitting. The small difference suggests that the model's complexity is balanced, and adding more variables may not drastically improve the explanation of variance.

Second, for LN: R-squared is 0.509, indicating that only 50.9% of the variance in LN is explained by the independent variables, which are relatively low. This suggests that the model has limited explanatory power for LN. While its adjusted

**Fig 2. Structural equation modeling (PLS- SEM).**

**Table 4. Inner VIF values.**

|  | VIF |
|---|---|
| BR→HVM | 1.179 |
| BR→LN | 1.048 |
| BR→TD | 1.048 |
| LN→HVM | 1.996 |
| LN→YD | 1.294 |
| TD→YD | 1.294 |
| UGC→HVM | 2.253 |
| UGC→LN | 1.132 |
| UGC→TD | 1.132 |
| YD→HVM | 2.358 |
| BR x UGC→LN | 1.094 |
| BR x UGC→TD | 1.094 |
| BR x YD→HVM | 1.321 |

*Notes:* BR is brand reputation; HVM is purchasing behavior for environmentally friendly products; LN is Environmental concerns; TD is Environmental Attitude; UGC is User-generated content; YD is Intention to purchase environmentally friendly products.

**Table 5. R-square and R-square adjusted.**

|  | R-square | R-square adjusted |
|---|---|---|
| HVM | 0.452 | 0.443 |
| LN | 0.509 | 0.505 |
| TD | 0.283 | 0.276 |
| YD | 0.395 | 0.392 |

R-squared is 0.505, showing a minimal difference from the R-squared. This small gap suggests that the inclusion of additional predictors would add little value in improving the model's performance for LN.

Third, for TD: R-squared is 0.283, meaning 28.3% of the variance in TD is explained by the independent variables. This value is moderate but still indicates that the model explains only a small portion of the variance in TD. Also, the adjusted R-squared is 0.276, closely aligning with the R-squared value, suggesting a reasonable model fit without much risk of overfitting.

Finally, for YD: R-squared is 0.395, meaning that 39.5% of the variance in YD is explained by the independent variables. Like TD, this value shows a modest explanatory capacity of the model. Besides, the adjusted R-squared is 0.392, again very close to the R-squared value, indicating that the model's fit is adequate, and there is little concern about overfitting.

According to the Cohen [25]'s rule of thumbs, $f^2 \geq 0.02$, $f^2 \geq 0.15$, and $f^2 \geq 0.35$ represent small, medium (moderate), and large effect sizes, respectively. From Table 6, we can conclude that:

First, strongest relationship: The impact of UGC on LN (0.649, large effect) is the most significant. This suggests that user-generated content plays a crucial role in influencing LN, potentially through eWOM or online discussions.

Second, moderate impacts: The relationships between LN and YD (0.208) and UGC and TD (0.168) indicate that these factors play important roles in shaping purchase behaviors.

Third, small effects: Many of the brand reputation (BR) effects are small, suggesting that while BR influences the model, it does not dominate the decision-making process. However, its interaction effects (BR × UGC and BR × YD) still hold some significance.

**Table 6. f square (f2).**

|  | f-square | Effect size |
|---|---|---|
| BR→HVM | 0.114 | Small |
| BR→LN | 0.044 | Small |
| BR→TD | 0.049 | Small |
| LN→HVM | 0.022 | Small |
| LN→YD | 0.208 | Medium |
| TD→YD | 0.137 | Medium |
| UGC→HVM | 0.026 | Small |
| UGC→LN | 0.649 | Large |
| UGC→TD | 0.168 | Medium |
| YD→HVM | 0.086 | Small |
| BR x UGC→LN | 0.063 | Small |
| BR x UGC→TD | 0.056 | Small |
| BR x YD→HVM | 0.114 | Small |

*Notes*: BR is brand reputation; HVM is purchasing behavior for environmentally friendly products; LN is Environmental concerns; TD is Environmental Attitude; UGC is User-generated content; YD is Intention to purchase environmentally friendly products.

Finally, minimal impact paths: The effects of UGC→HVM (0.026) and LN→HVM (0.022) suggest these relationships are weak and may require further investigation or additional influencing factors.

## Discussions

Based on hypothesis testing results, these relationships are then assessed through the coefficients, t-statistics, and P-values obtained in Table 7.

The results show that, first, UGC directly affects LN: a 0.601 positive coefficient shows that increased levels of user-generated content are associated with significant increases in environmental concerns. This association is even stronger with a T-statistic of 11.125 (higher than the p-value of 0.000 also confirming the reliability of this relationship, proving that means UGC is a predictor of means LN and this variable highly explained in the model, enabling a more reliable hypothesis 1 (Table 7). These results match those from [91,103,138]. Restaurant chains are usually less environmentally sustainable [76,108] due to the tremendous environmental impact [62] of which they generated large amounts of food and plastic waste and emissions as well as consumed excess water and energy.

Second, the path of UGC to environmental attitude (TD) at Hospitality and Food Service Industry in Vietnam, especially in Ho Chi Minh City shows a significantly positive relationship, with a coefficient of 0.370. As UGC increases, TD experiences a considerable rise, making this the most pronounced effect among the relationships studied. The T-statistics of 5.896 further emphasize the statistical significance of this relationship, being well above the critical value of 1.96, signifying both strength and reliability. Additionally, the P-value of 0.000 confirms the significance, indicating that the probability of this result occurring by chance is virtually zero. In addition, hypothesis 2 is confirmed in the above analysis, which means that UGC positively influences environmental attitudes at Hospitality and Food Service Industry in Vietnam, especially in Ho Chi Minh City, as demonstrated by [61,120,138] who demonstrate that consumers use UGC to assess the environmental aspects of a restaurant, such as the use of eco-friendly products or sustainable practices. By contributing feedback and sharing reviews about these experiences, UGC helps build the restaurant's reputation for sustainability. This reinforces consumers' positive attitudes toward the restaurant and influences their purchasing behavior toward environmentally friendly products and services. On the platform of the [120] study, UGC fosters positive environmental attitudes

**Table 7. Hypothesis testing.**

| | Original sample (O) | Sample mean (M) | Standard deviation (STDEV) | T statistics (|O/STDEV|) | P values |
|---|---|---|---|---|---|
| BR→HVM | 0.271 | 0.269 | 0.088 | 3.079 | 0.002 |
| BR→LN | 0.151 | 0.154 | 0.038 | 3.942 | 0.000 |
| BR→TD | 0.192 | 0.188 | 0.053 | 3.637 | 0.000 |
| LN→HVM | 0.154 | 0.151 | 0.066 | 2.349 | 0.019 |
| LN→YD | 0.403 | 0.404 | 0.056 | 7.187 | 0.000 |
| TD→YD | 0.327 | 0.326 | 0.054 | 6.007 | 0.000 |
| UGC→HVM | 0.180 | 0.175 | 0.076 | 2.371 | 0.018 |
| UGC→LN | 0.601 | 0.601 | 0.054 | 11.125 | 0.000 |
| UGC→TD | 0.370 | 0.367 | 0.063 | 5.896 | 0.000 |
| YD→HVM | 0.334 | 0.341 | 0.091 | 3.681 | 0.000 |
| BR x UGC→LN | −0.133 | −0.136 | 0.033 | 4.047 | 0.000 |
| BR x UGC→TD | −0.151 | −0.140 | 0.051 | 2.936 | 0.003 |
| BR x YD→HVM | 0.168 | 0.168 | 0.058 | 2.890 | 0.004 |

*Notes*: BR is brand reputation; HVM is purchasing behavior for environmentally friendly products; LN is Environmental concerns; TD is Environmental Attitude; UGC is User-generated content; YD is Intention to purchase environmentally friendly products.

by promoting emotional engagement with environmental issues. Greater exposure to environmentally focused UGC aligns individual attitudes with responsible environmental behavior.

Third, the path of UGC to HVM shown in Table 7 indicates a small but positive impact, with a coefficient of 0.180. This suggests that while UGC contributes to changes in HVM, its effect is relatively modest. Nonetheless, the influence is positive, meaning that as UGC increases, HVM also increases, though to a lesser extent compared to other variables. The T-statistics of 2.371, being slightly above the critical threshold of 1.96, indicate that this relationship is statistically significant. Although the strength of the effect is not substantial, it is reliable enough to be considered meaningful. The p-value of 0.018 further supports the conclusion that UGC has a statistically significant influence on HVM. While the effect size may be small, the significance of this relationship confirms that UGC plays a notable, albeit modest, role in affecting HVM within the model. Thus, hypothesis 3 is confirmed because of the significant p-value, which leads to the positive and significant effect of UGC on Purchasing Behavior for Environmentally Friendly Products. The results are aligned to Bahtar & Muda [6]; Horst et al. [55]; Racherla & Friske [103]; Yan et al. [138].

Considering the path between LN and HVM (purchase behavior), the relationship between LN and HVM demonstrates a moderate positive effect, with a path coefficient of 0.154. Based on the path coefficient, the results indicate that LN directly influences HVM. Besides, the T-statistic of 2.349, exceeding the critical value of 1.96, confirms statistical significance, with a very low likelihood of the result occurring by chance. Additionally, the p-value of 0.019 reinforces the reliability of this relationship, highlighting LN's critical role in shaping HVM within the model. Also, the evidence representing hypothesis 5 confirms that environmental concerns significantly influence purchasing environmentally friendly products at Hospitality and Food Service Industry in Vietnam, as mentioned by Balaskas et al. [7]; Kim & Choi [64]; Mostafa [86]; Yadav & Pathak [137].

The next relationship between YD and HVM demonstrates a moderate positive effect, with a coefficient of 0.334, indicating that YD meaningfully influences HVM, as increases in YD lead to moderate increases in HVM. The T-statistics of 3.681, significantly above the critical value of 1.96, confirms the statistical significance of this relationship, suggesting that the observed effect is reliable and unlikely to result from random chance. Moreover, the P-value of 0.000 further affirms the strength of this connection, establishing YD as a strong and significant predictor of HVM. These findings confirm that YD plays a crucial role in driving changes in HVM within the model. As analyzed, hypothesis 6 is validated, showing that the purchase intention for environmentally friendly products positively influences purchasing behavior for such products at 5-star restaurants in Ho Chi Minh City. This relationship is confirmed by [21], Hassan et al. [48], 71, Thi Tuyet et al. [124].

Significantly, the findings confirm that brand reputation moderates the relationship between UGC and environmental concern; UGC and environmental attitude; intention to purchase environmentally friendly products and purchasing behavior.

The findings indicate that brand reputation negatively moderates the relationship between UGC and environmental concern (LN) (Effect size: −0.133, T-statistic: 4.047, p = 0.000). This suggests that consumers rely less on UGC for environmental concerns when engaging with reputable brands, whereas lesser-known brands benefit more from UGC-driven consumer awareness. The results align with Signaling Theory, which posits that brand reputation reduces information asymmetry, making consumers less dependent on peer-generated content [12,28]. Similarly, Brand-Trust Theory highlights that strong brands enhance consumer confidence in sustainability claims, diminishing UGC's influence [56]. For lesser-known brands, UGC plays a vital role in establishing credibility. Engaging sustainability influencers and promoting consumer reviews can enhance trust. Meanwhile, well-established brands should reinforce environmental credibility through corporate sustainability initiatives. This study contributes to sustainability marketing by clarifying how brand reputation shapes consumer reliance on UGC for environmental awareness. These findings contribute to the broader literature on consumer behavior and sustainability marketing by clarifying how brand reputation conditions the effect of UGC on environmental concern. Future research could further investigate how different types of UGC (e.g., video reviews, blog articles, or expert testimonials) interact with brand reputation in shaping consumer attitudes toward environmental sustainability.

The findings indicate that brand reputation negatively moderates the impact of user-generated content (UGC) on environmental attitudes (Effect size: −0.151, T-statistic: 2.936, p = 0.003). This suggests that for well-established brands, consumers rely more on brand credibility than UGC when forming environmental attitudes. However, for lesser-known brands, UGC plays a more significant role in shaping consumer perceptions. This aligns with Signaling Theory, which posits that brand reputation serves as a credibility signal, reducing consumers' reliance on external information sources like UGC [28]. Additionally, Brand-Trust Theory suggests that trust mitigates perceived risks, further weakening the influence of UGC when a brand's reputation is strong [37]. Unlike previous studies [1,126], which emphasized UGC's universal influence on environmental attitudes, our results highlight that this effect is conditional on brand reputation. In Vietnam's hospitality sector, this means that strong brands should reinforce sustainability messaging directly, while lesser-known brands can leverage UGC-driven strategies to build credibility.

The results show that brand reputation positively moderates the relationship between purchase intention (YD) and actual purchasing behavior (HVM) (Effect size: 0.168, T-statistic: 2.890, p = 0.004), meaning that stronger brand reputation increases the likelihood of consumers acting on their purchase intentions. These findings align with Expectancy-Value Theory, where brand reputation serves as a signal of trust and authenticity, reducing perceived risks and reinforcing consumer confidence. Empirical studies [67,88] confirm that reputable brands facilitate purchase decisions, while skepticism towards weaker brands hinders conversions [69]. For Vietnam's hospitality industry, strong brands should leverage trust to promote eco-friendly products, while emerging brands must focus on transparency and engagement. This study contributes to Brand-Trust Theory [84] and sustainability marketing by highlighting how reputation strengthens the intention-behavior link.

## Conclusion and implications

### Conclusions

The quantitative research methods have been utilized to meet the study's objectives. This study began by examining the factors of UGC that determine environmentally friendly purchase behavior in the 5-star restaurant setting of Ho Chi Minh City. UGC significantly affects environmental concerns, attitudes towards it, and the intention to buy green products. The study also analyses the power of each factor in derived UGC on green product purchase intention at Hospitality and Food Service Industry in Vietnam, especially in Ho Chi Minh City. The findings confirm that brand reputation moderates the impact of user-generated content (UGC) on environmental concern, environmental attitudes, and purchasing behavior. The findings confirm that brand reputation is a key moderator in consumer decisions on environmentally friendly products. Consumers of well-established brands rely more on brand credibility and less on UGC, while lesser-known brands benefit more from UGC-driven trust-building. Strong brand reputation also increases the likelihood of converting purchase intentions into actual behavior. These insights contribute to sustainability marketing, showing that UGC's influence depends on brand reputation. Businesses should tailor their strategies, strong brands should reinforce credibility through direct messaging, while emerging brands should leverage UGC to build trust. Future research could explore how different UGC formats interact with brand reputation in shaping consumer perceptions.

### Theoretical implications and practical implications

This study extends the application of consumer behavior theories, including the TPB, Signaling Theory, and Brand-Trust Theory, in the context of the hotel and food service industry in Vietnam. The results show that UGC not only directly impacts environmental awareness and attitudes, but also indirectly promotes environmentally friendly purchase intentions and behavior. In particular, brand reputation is identified as an important moderator, helping to explain the difference in the level of consumer reliance on UGC between strong and weak brands. This finding contributes to strengthening the Signaling Theory by demonstrating that a high brand reputation helps reduce information asymmetry and mitigates the impact of UGC. At the same time, the study also complemented the Brand Trust Theory by demonstrating that brand trust and reputation can strengthen the link between green product purchase intention and behavior.

In practical, the research results provide valuable insights for developing sustainable marketing strategies in the hotel and catering industry. Major brands need to continue to maintain and strengthen their green image through official communication on environmental responsibility to increase customer trust. Meanwhile, less established brands should effectively leverage UGC, such as customer reviews, comments, and shares, to enhance their reputation and build trust. In addition, cooperating with environmental influencers or encouraging customers to post green reviews are also practical solutions to increase trust, form positive attitudes, and promote the intention to buy environmentally friendly products.

In addition, the research results also suggest some specific strategic directions to help businesses effectively exploit UGC to enhance green brand image and promote positive shopping behavior. Specifically, businesses can build online platforms that encourage customers to share green consumption experiences, while implementing proactive feedback mechanisms to increase engagement. In addition, integrating UGC content into brand communication strategies – such as citing real reviews in promotional campaigns or organizing "UGC campaigns" to spread sustainable values – will help strengthen credibility and increase consumer trust. Finally, businesses should combine UGC with customer behavior data to analyze green consumption trends, thereby adjusting marketing strategies more flexibly and effectively.

### Limitations and future research

The study was conducted at 5-star restaurants in Ho Chi Minh City, so the results may not fully reflect other market segments or international contexts. Future studies can expand the scope of the survey to popular restaurant groups or food service chains in other areas to test the generalizability of the model. Furthermore, the various forms of UGC, such as video reviews, blogs, or influencer content, should be further considered to assess differences in their influence on consumer perceptions and behaviors. In addition, integrating additional variables such as "green trust", "green perceived value", or "subjective norm" into the model will help expand and strengthen the TPB theoretical framework, contributing to clarifying the impact mechanism of UGC on the purchasing behavior of environmentally friendly products.

## Author contributions

**Formal analysis:** Tu Ngoc Tran.

**Investigation:** Tu Ngoc Tran.

**Methodology:** Tu Ngoc Tran.

**Project administration:** Tu Ngoc Tran.

**Resources:** Nguyen Phan Thu Hang.

**Software:** Nguyen Phan Thu Hang.

**Supervision:** Tu Ngoc Tran.

**Validation:** Nguyen Phan Thu Hang.

**Visualization:** Nguyen Phan Thu Hang.

**Writing – original draft:** Tu Ngoc Tran, Nguyen Phan Thu Hang.

**Writing – review & editing:** Tu Ngoc Tran.

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
