## [Decision Letter · Decision Letter 0]

28 Sep 2025

Dear Dr. Tran,

We look forward to receiving your revised manuscript.

Kind regards,

Jenna Scaramanga

Staff Editor

PLOS ONE

Journal Requirements:

5. Please upload a new copy of Figure 2 as the detail is not clear. Please follow the link for more information: https://blogs.plos.org/plos/2019/06/looking-good-tips-for-creating-your-plos-figures-graphics/

Reviewers' comments:

Reviewer's Responses to Questions

**Comments to the Author**

1. Is the manuscript technically sound, and do the data support the conclusions?

Reviewer #1: Yes

Reviewer #2: Yes

2. Has the statistical analysis been performed appropriately and rigorously?

Reviewer #1: Yes

Reviewer #2: Yes

3. Have the authors made all data underlying the findings in their manuscript fully available?

Reviewer #1: Yes

Reviewer #2: No

4. Is the manuscript presented in an intelligible fashion and written in standard English?

Reviewer #1: Yes

Reviewer #2: Yes

Reviewer #1: This article aims to explore the effect of User-Generated Content (UGC) on the purchasing behavior of environmentally friendly products in the Hospitality and Food Service Industry. It is an interesting and valuable field of research. However, how can the author say that this study uses a strong foundation, especially regarding the TPB? The author only applies one construct from the TPB, namely attitude. Furthermore, the implications of the study are not clearly presented, both theoretically and practically.

There is a lack of adequate foundation in the literature, and the definition of basic concepts is neglected. In addition, previous research supporting the hypothesis is also inadequate.

The methods section is well-designed, but the rationale for choosing PLS-SEM over other software for this study needs to be clarified.

The results of the study are presented clearly and logically, and the analysis is well-grounded in the methodology used. The article effectively demonstrates how UGC influences the environmental concerns, attitudes, and the intention to purchase green products. As well as the moderating role of brand reputation. Authors can add notes in each table for variable abbreviations. The conclusions are consistent with the earlier parts of the article and effectively combine theory with empirical data. The only possible improvement could be a deeper comparison of the results with previous studies in other cultural contexts.

The findings have direct implications only for businesses, emphasizing the importance of UGC to build trust regarding environmentally friendly products in the Hospitality and Food Service Industry. It is recommended to create new sections for each implication, both theoretical and practical, including limitations and suggestions for future research.

Reviewer #2: The document considers the effect UGC has on the purchase behavior in hospitality and food service industries while taking into account the moderation effect of brand reputation. A relevant model has successfully incorporated the framework with relevant theories and empirical research. The results affirm the hypotheses, and the conclusions drawn enhance the knowledge of consumer behavior toward sustainable consumption. In my opinion though, the research can be improved in the research design and practical use aspects.

1. Availability of the Data: Athough the authors mention the availibility of the data, it is available only by request, which is contrary to the policy of unrestricted data availibility of PLOS ONE. Suggestion: The authors must either upload the data to a public repository, or provide a reasonable justification for their inability to provide the data in full.

2. Sample Representation: Though a sample of 331 participants is sizable, drawing all participants exclusively from five-star dining establishments in Ho Chi Minh City may be too limiting to reflect the entire consumer base in Vietnam. Suggestion: Including participants from a wider array of dining establishments and across different geographic areas in Vietnam would enhance the applicability and usefulness of the results.

3. Practical Implication: While the focus in the manuscript on the theoretical part is pretty reasonable, it is the practical approach in relation to the business strategy which seems to be missing in this case. Suggestion: The authors must come up with strategies focused on the use of UGC to elevate brand image and positively influence consumer buying behaviour on green products, as well as practical strategies for the business.

4. Theoretical Framework and Literature Review: Having a strong theoretical foundation is great, but the literature review needs more explanation on the application of such theories on the hospitality market in the like of Vietnam. Recommendation: The authors should add additional literature that has analyzed the impact of UGC on consumer behavior in the hospitality industry in emerging economies.

5. Hypothesis Development: The authors did a great job in formulating the hypotheses but additional justification is a must for why the brand reputation is a strong moderator on the impact of UGC toward consumer behavior in the hospitality sector. Recommendation: The authors should explain in more detail why UGC is likely to have more impact on purchasing behavior on high reputation brands compared to low reputation brands.

**Do you want your identity to be public for this peer review?** For information about this choice, including consent withdrawal, please see our Privacy Policy

Reviewer #1: No

Reviewer #2: **Yes: ** Dicky Jhon Anderson Butarbutar

---

## [Author Response · Author response to Decision Letter 1]

14 Oct 2025

Dear Sir/Madam,

We have revised the manuscript and submitted it online.

We have also attached the file containing our responses to the reviewers.

Thank you and regards,

Ngoc Tu

---

## [Decision Letter · Decision Letter 1]

6 Nov 2025

Research on the impact of User-Generated Content (UGC) in shaping the purchase behavior of environmentally friendly products and the moderating role of brand reputation

PONE-D-25-13451R1

Dear Dr. Tran,

We’re pleased to inform you that your manuscript has been judged scientifically suitable for publication and will be formally accepted for publication once it meets all outstanding technical requirements.

Kind regards,

Vanessa Carels

Staff Editor

PLOS ONE

Additional Editor Comments (optional):

Reviewers' comments:

Reviewer's Responses to Questions

**Comments to the Author**

Reviewer #1: All comments have been addressed

2. Is the manuscript technically sound, and do the data support the conclusions?

Reviewer #1: Yes

3. Has the statistical analysis been performed appropriately and rigorously?

Reviewer #1: Yes

4. Have the authors made all data underlying the findings in their manuscript fully available?

Reviewer #1: Yes

5. Is the manuscript presented in an intelligible fashion and written in standard English?

Reviewer #1: Yes

Reviewer #1: Overall, you did a good job revising the paper.

In the background, you have strongly explained the reason for choosing one construct from TPB, namely attitude.

The methods section is well-designed; you already clarified the rationale for choosing PLS-SEM over other software for this study.

The results of the study are presented clearly and logically, and the analysis is well-grounded in the methodology used. The article effectively demonstrates how UGC influences the environmental concerns, attitudes, and the intention to purchase green products, in addition to the moderating role of brand reputation.

The authors have already added notes in each table for variable abbreviations. The conclusions are consistent with the earlier parts of the article and effectively combine theory with empirical data.

The authors already created new sections for each implication, both theoretical and practical, including limitations and suggestions for future research. It is essential to enhance the robustness of this article.

**Do you want your identity to be public for this peer review?** For information about this choice, including consent withdrawal, please see our Privacy Policy

Reviewer #1: **Yes: ** Prof. Dr. Ratni Prima Lita, SE, MM

---

## [Editor Report · Acceptance letter]

PONE-D-25-13451R1

PLOS ONE

Dear Dr. Tran,

I'm pleased to inform you that your manuscript has been deemed suitable for publication in PLOS ONE. Congratulations! Your manuscript is now being handed over to our production team.

Kind regards,

on behalf of

Dr. Vanessa Carels

Staff Editor

PLOS ONE